# Loss of DNA Polymerase β Delays Atherosclerosis in *ApoE*^−/−^ Mice Due to Inhibition of Vascular Smooth Muscle Cell Migration

**DOI:** 10.3390/ijms252111778

**Published:** 2024-11-02

**Authors:** Lianfeng Zhao, Jiannan Chen, Yan Zhang, Jiaqi Liu, Wenying Li, Yuling Sun, Ge Chen, Zhigang Guo, Lili Gu

**Affiliations:** Jiangsu Key Laboratory for Molecular and Medical Biotechnology, College of Life Sciences, Nanjing Normal University, 1 Wenyuan Road, Nanjing 210023, China; zlf365718@163.com (L.Z.); cjn.njnu@foxmail.com (J.C.); z1265348898@163.com (Y.Z.); thirchyljq@163.com (J.L.); liwenying0223@163.com (W.L.); yulingsun@szhct.edu.cn (Y.S.); cg19971001@163.com (G.C.); guozgang@gmail.com (Z.G.)

**Keywords:** Pol β, atherosclerosis, vascular smooth muscle cell, migration, YY1/TGF-β1 pathway

## Abstract

Atherosclerosis (AS) is an inflammatory disease characterized by arterial inflammation. One important trigger for AS development is the excessive migration of vascular smooth muscle cells (VSMCs); however, the mechanism underlying this phenomenon remains unclear. Therefore, we investigated the role of DNA polymerase β (Pol β), a crucial enzyme involved in base excision repair, VSMC migration, and subsequent AS development. In this study, we revealed a significant increase in Pol β content within AS plaques in *ApoE*^−/−^*Pol β*^+/+^ mice. In vitro experiments demonstrated a significant decrease in hCASMC viability and migration ability upon Pol β knockdown, whereas the subsequent recovery of Pol β expression reversed this effect. Moreover, our investigations revealed that Pol β knockdown leads to the inhibition of the *POSTN* gene transcription by suppressing the YY1/TGF-β1 pathway, resulting in the decreased expression of the protein periostin during VSMC migration. Collectively, our findings provide insights into the role of Pol β in AS development, offering a novel approach for the clinical treatment of cardiovascular diseases.

## 1. Introduction

Atherosclerosis (AS) is a prevalent intrinsic manifestation of cardiovascular disease, primarily characterized by arterial inflammation driven by small lipid molecules [1,2]. AS lesions arise from lipid deposition within the arterial walls, which is accompanied by the hyperplasia of vascular smooth muscle cells (VSMCs), eventually leading to the formation of atherosclerotic plaques. Initially, research focused solely on lipid accumulation [3], the release of inflammatory factors [4,5], the involvement of immune cells [6], and the transformation of macrophages into foam cells [7,8]. However, the role of VSMCs in this process is gaining increasing attention with further investigations [9,10]. Some VSMCs undergo phenotypic conversion into macrophage-like VSMCs, leading to lipid uptake and the formation of foam cells, which contribute to secondary necrosis and inflammation [11]. Additionally, VSMCs produce macrophage colony-stimulating factor (M-CSF), promoting the proliferation of macrophages within atherosclerotic lesions [12]. The excessive migration of VSMCs further exacerbates the progression of AS [13]. Nevertheless, the precise mechanisms underlying the acceleration of AS by VSMCs remain unclear.

Periostin is a disulfide-linked protein that is secreted by osteoblasts and osteoblast-like cell lines and has a molecular weight of 93.3 kDa [14]. In the context of myocardial fibrosis, periostin plays a role in influencing myocardial fibroblasts and promoting extracellular matrix (ECM) synthesis, thereby affecting collagen production. Periostin is a downstream target of transforming growth factor-β1 (TGF-β1), which acts on cardiac fibroblasts (CFs) and cardiomyocytes [15]. Recent research has revealed that the zinc-fingered transcription factor Ying Yang 1(YY1) functions as a potent transcriptional repressor of TGF-β1 by binding to its DNA regulatory regions [16]. Furthermore, the activity of YY1 is associated with the expression of periostin [17]. Additionally, Twist1 has been identified as a positive regulator of periostin expression [18,19,20,21], and TGF-β1 has been shown to promote collagen synthesis and smooth muscle cell proliferation [22].

Base excision repair (BER) is a critical mechanism within the DNA damage repair pathway [23], with Pol β serving as the pivotal repair enzyme [24]. DNA damage impedes the excessive migration of VSMCs [25], suggesting a correlation between Pol β and the development of AS disease. Additionally, DNA damage has been implicated in the promotion of advanced AS plaque shedding and cell death [26]. Shah et al. further demonstrated that DNA damage resulting from oxidative stress exacerbates AS progression [25].

Accordingly, our study aimed to clarify the critical role of Pol β in the pathogenesis of AS. Overall, our findings highlight the importance of Pol β in AS development and lay the groundwork for understanding the role of DNA damage repair mechanisms in the pathophysiology of cardiovascular diseases.

## 2. Results

### 2.1. Pol β Knockdown Inhibits the Formation of AS Plaques in ApoE^−/−^ Mice

To investigate whether Pol β is involved in the occurrence of AS, mice were fed a high-fat diet to induce the development of AS. Noticeable aortic plaques were observed after 6, 12, and 18 weeks of the high-fat diet, confirming the successful establishment of the AS model (Figure 1A,B). Meanwhile, we measured Pol β levels in AS plaques and found that Pol β was significantly up-regulated in the plaques of *ApoE*^−/−^ mice, whereas no significant difference was observed in the hearts of *ApoE*^−/−^ mice (Figure 1C–F). These results indicated that Pol β may be associated with AS occurrence. To further clarify the relationship between Pol β and AS development, we crossed the mice for at least three generations and established *ApoE*^−/−^*Pol β^+/^*^−^ mice. To further clarify the role of Pol β in AS, we developed transgenic mouse models. Our findings revealed a significant reduction in AS severity in *ApoE*^−/−^*Pol β*^+/−^ mice (Figure 1G–I). α-SMA acts as a biomarker for VSMCs [27]: the levels of α-SMA in the VSMCs of *ApoE*^−/−^*Pol β*^+/−^ mice were markedly decreased compared to those in *ApoE*^−/−^*Pol β*^+/+^ mice, as shown by immunofluorescence assays (Figure 1J,K). Bedford et al. identified Pol β as a DNA damage repair enzyme that is also involved in the protein arginine process [28,29,30,31] and the regulation of demethylation [32]. Therefore, we hypothesized that Pol β may possess other unexplored auxiliary effects in AS development in *ApoE*^−/−^ mice.

### 2.2. Pol β Down-Regulation Inhibits VSMC Migration

VSMC migration plays an important role in AS development. Therefore, we examined whether Pol β affects AS by affecting VSMC migration. Our data demonstrated that cell migration was significantly reduced in Pol β-knockdown hCASMC (Figure 2A,B), whereas reverse Pol β-knockdown hCASMCs partially recovered the inhibition of migration (Figure 2C,D). We further assessed the expression levels of the migration-related proteins, N-cadherin, and E-cadherin. In Pol β-knockdown hCASMCs, we observed a decrease in both Vimentin and N-cadherin, while E-cadherin levels were increased. However, the regulatory effects of these proteins diminished following the restoration of Pol β expression (Figure 2E,F). These findings collectively indicated that the migration capacity was notably impeded in Pol β-knockdown hCASMCs. Additionally, the transwell experiment further confirmed that knocking down Pol β significantly inhibited the migration ability of hCASMCs (Figure 2G). Moreover, cell viability was significantly reduced in Pol β knockdown-hCASMCs, as measured by the CCK8 assay (Figure 2H). Migration ability and cell viability were restored in reverse Pol β-knockdown hCASMCs (Figure 2G,H). Taken together, these data indicated that Pol β was involved in the proliferation and migration of VSMCs.

### 2.3. Pol β Down-Regulation Inhibits the Migration of VSMCs via Periostin

We previously conducted RNAseq experiments in MEFs with knocked-out *Pol β*: the results showed that Pol β deficiency led to a decrease in multiple cytoadhesin molecules and certain cytoplasmic matrix molecules, particularly periostin, which showed the most significant change in *Pol β*^−/−^ MEF cells (Figure 3A) [33]. Furthermore, qRT-PCR verification was performed on MEF cells, and our data confirmed that in addition to collagen *Col6α2*, the mRNA levels of *Postn*, *Col1α1*, *Col3α1*, and *Col6α1* were significantly reduced in *Pol β*^−/−^ MEF cells (Figure 3B,C). A similar effect of periostin was observed when *Pol β* down-regulation with siRNA was investigated in hCASMCs (Figure 3D,E). The loss of periostin in a mouse AS model decreased mature collagen production and inflammatory cell activation and migration and altered VSMC behavior [34]. After confirming the effect of Pol β on periostin, we investigated whether Pol β is involved in the production of the extracellular matrix. As shown in Figure 3F,G, *COL1α1*, *COL3α1*, *COL5α1*, and *MMP9* were significantly increased, whereas *MMP3* and *MMP13* were significantly reduced in Pol β-knockdown hCASMCs and *MMP2* expression remained unchanged. Conversely, the up-regulation of collagens induced by Pol β knockout was abolished upon restoration of Pol β expression, whereas *MMP2*, *MMP3*, *MMP9*, and *MMP13* expressions were up-regulated. These findings suggest that Pol β expression correlates with periostin and the extracellular matrix.

To further clarify whether Pol β affects VSMC migration via periostin, we first confirmed that *POSTN* knockdown significantly decreased hCASMC migration, whereas the restoration of Pol β expression abolished these effects (Figure 3H,I). Next, we overexpressed *POSTN* after knocking down Pol β to detect cell migration. The expression of Vimentin and N-cadherin were dramatically increased, while E-cadherin expression was markedly decreased compared to the Pol β knockdown group (Figure 3J). Transwell experiments and CCK8 assays demonstrated that the migration ability and cell viability of hCASMCs were restored to wild-type (WT) levels after Pol β knockdown followed by *POSTN* overexpression (Figure 3K,L). These findings indicated that the overexpression of *POSTN* could counteract the reduced cell migration ability and viability caused by Pol β knockdown. Additionally, the levels of *POSTN* in the VSMCs of *ApoE*^−/−^*Pol β^+/−^* mice were markedly decreased compared to those in *ApoE*^−/−^*Pol β*^+/+^ mice, as shown by immunofluorescence assays (Figure 3M). Collectively, our results provide clear evidence that Pol β is involved in the migration of VSMCs via periostin.

### 2.4. Pol β Knockdown Inhibits the Expression of Periostin Through the YY1-TGF-β1 Pathway

*YY1* was previously demonstrated to be a transcriptional inhibitor of *POSTN* [17]. Furthermore, YY1 can directly bind to TGF-β1 promoter regions and repress TGF-β1 transcription [16]. TGF-β1 can promote POSTN expression and secretion by prompting its downstream gene, Smad3, to directly bind to the *POSTN* promoter region [35]. After confirming the effect of Pol β on periostin, we investigated whether the YY1/TGF-β1 pathway was involved in the molecular mechanisms of this effect.

The mRNA expression levels of *YY1*, *TGF-β1*, and *TWIST1* in hCASMCs with knocked-down Pol β were measured using qRT-PCR. The transcription of *TGF-β1* and periostin were found to be markedly decreased, whereas *TWIST1* and *YY1* showed no significant difference after Pol β knockdown. However, a slight increase in YY1 transcription was observed (Figure 4A). These results indicate that the transcriptional decrease of periostin is not regulated by the transcription factor *TWIST1* in Pol β-knockout hCASMCs. Furthermore, we found that YY1 protein levels were up-regulated, while TGF-β1 levels were down-regulated in Pol β-knockdown VSMCs (Figure 4B,C). However, the expression of YY1 was down-regulated and those of TGF-β1 and periostin were up-regulated upon re-expression of Pol β (Figure 4B,C). Subsequently, the down-regulation of periostin was also abolished by knocking down YY1 or SRI-011381 (a TGF-β1 agonist) pretreatment in Pol β-knockdown hCASMCs (Figure 4D–G). It was proved that Pol β knockdown was significantly down-regulated periostin by YY1/TGF-β1 axis in hCASMCs.

We further investigated whether the molecular mechanism by which Pol β knockdown inhibits periostin expression is related to the YY1/TGF-β1 pathway in AS mice. The results showed that the levels of TGF-β1 were notably up-regulated, whereas those of YY1 were down-regulated in the AS lesions of *ApoE*^−/−^*Pol β*^+/+^ mice. These levels could be abolished by Pol β deficiency, as shown by immunofluorescence assays (Figure 4H,I). Overall, our results provide clear evidence that Pol β is involved in AS through the YY1/TGF-β1 pathway in vivo and in vitro.

## 3. Discussion

This study confirms the involvement of Pol β in the initiation and progression of AS diseases. The inhibition of Pol β expression was found to alleviate AS diseases in *ApoE*^−/−^ mice. Pol β deficiency suppresses the YY1/TGF-β1 pathway and periostin expression, thereby impeding the migration of SMCs and inhibiting the progression of AS diseases (Figure 5).

Most of the existing studies on the relationship between DNA damage and AS have indicated that severe DNA damage is associated with AS development [26]. DNA damage is observed in the mitochondrial DNA genome and AS lesions [36]. Oxidative-stress-induced DNA damage has been identified in failing hearts and other cardiovascular diseases in the study conducted by Shukla [37]. Oxidative stress potentially leads to an up-regulation of Pol β to facilitate DNA repair. The enzyme OGG1 plays a protective role in atherogenesis by preventing the excessive activation of the inflammasome, and its deficiency can exacerbate AS progression [38]. Krokan et al. demonstrated that the base excision repair (BER) pathway is a key DNA damage repair pathway that plays an additional significant role in adaptive immunity and epigenetics [39]. OGG1 is a critical enzyme involved in the BER pathway, functioning as a repair enzyme and additionally participating in the development of neurodegenerative diseases [40] and the processes of methylation and demethylation [41]. Inflammatory processes in atherosclerosis, driven by factors such as dyslipidemia and endothelial dysfunction, can also affect Pol β expression. Pro-inflammatory cytokines may modulate the expression of DNA repair enzymes, including Pol β, as part of the cellular response to inflammation-induced DNA damage. Studies have shown that mutations in Pol β in mice can lead to dysplasia in early mouse embryos [42]. Therefore, Pol β is speculated to act as a regulatory protein that modulates the activity of transcription factors in *ApoE*^−/−^*Pol β*^+/+^. Some evidence suggests that DNA damage repair is associated with cardiovascular disease [43,44]; however, the specific mechanisms underlying this association are not fully understood. Further research is needed to elucidate the complex interplay between Pol β, DNA repair, and the various risk factors associated with atherosclerosis.

Our findings demonstrated that the down-regulation of Pol β can effectively delay the progression of AS in *ApoE*^−/−^*Pol β*^+/+^mice. Interestingly, we observed that *ApoE*^−/−^*Pol β*^+/−^mice exhibited a leaner phenotype compared to that of *ApoE*^−/−^*Pol β*^+/+^ mice. This may be attributed to impaired lipid synthesis with the *ApoE*^−/−^ background, resulting in the blocking of LDL conversion into VLDL. Notably, Pol β knockout effectively preserved this process, leading to the inhibition of AS progression. Moreover, the aortic root of *ApoE*^−/−^*Pol β*^+/−^mice displayed a significant reduction in plaque formation, accompanied by a relatively flat area surrounding the aortic root. These findings suggest a potential association between ApoE and Pol β, which warrants further investigation in future studies.

Excessive migration of VSMCs contributes to the development and progression of AS [45]. Li et al. identified periostin, a cytoplasmic matrix protein, as a key regulator of VSMC migration [46]. Periostin has also been shown to promote tissue remodeling and cardiac regeneration following myocardial infarction [47,48]. Its expression is regulated by TGF-β1. Inhibiting TGF-β1 and suppressing the expression of periostin significantly down-regulates the expression of other collagen molecules in cells [49], in line with the findings of this study.

In recent years, extensive research has been conducted to investigate the role of YY1 in various biological processes, including skeletal muscle differentiation [50], autophagy [51], tumor proliferation [52], VSMC proliferation [53], and diabetes [54], among others. Previous studies have reported that the YY1 protein can improve the pathology of diabetic nephropathy by inhibiting the transcription of TGF-β1 [16]. Romeo et al. demonstrated that the activity of the human *POSTN* gene promoter, which promotes transcription, is primarily dependent on the activity of YY1 [17]. Inhibiting Pol β leads to a significant down-regulation of periostin expression, a phenomenon associated with the YY1/TGF-β1 pathway. Previous research has indicated that Pol β is involved in protein methylation and demethylation processes. Pol β can demethylate the DNA of the CDH13 promoter, thereby promoting the expression of CDH13 and facilitating cell adhesion and migration functions, which are involved in the regulatory processes of breast and lung cancer [33]. This finding supports the conclusion of the present study. Furthermore, our study revealed that Pol β knockdown results in a significant increase in YY1 expression, suggesting that Pol β may be involved in the transcriptional regulation of YY1 through the methylation process. However, further investigations are required to fully understand the interaction between Pol β and YY1.

In conclusion, we unveiled a novel role of Pol β in the pathogenesis of AS. Deficiency of Pol β leads to an up-regulation of YY1 expression and down-regulation of TGF-β1 expression, consequently inhibiting periostin, which reduces the levels of cytoplasmic matrix proteins and hinders the migration of VSMCs, ultimately mitigating the progression of AS. Pol β regulates AS by inhibiting VSMC migration via the YY1/TGF-β1 pathway in *ApoE*^−/−^ mice. These findings provide compelling evidence for the involvement of Pol β in AS in *ApoE*^−/−^ mice and suggest that targeting Pol β may be an effective strategy for the prevention and treatment of AS and other related cardiovascular diseases. Notably, this study is limited to *ApoE*^−/−^ mice and does not investigate clinical outcomes. Additionally, numerous additional protein molecules may influence the excessive migration of SMC. Further investigations are warranted to explore these relationships in the future.

## 4. Materials and Methods

### 4.1. Reagents and Antibodies

Dulbecco’s modified eagle medium (DMEM), RPMI 1640, fetal bovine serum (FBS), and Opti-MEM medium were purchased from Thermo Fisher Scientific (Waltham, MA, USA). Antibodies against Pol β, Vimentin, and YY1 were obtained from Abcam (Cambridge, UK). Antibodies against N-cadherin was obtained from Proteintech (Wuhan, China). Antibodies against β-tubulin, GAPDH, E-cadherin, α-SMA, and TGF-β1 were obtained from Abclonal (Wuhan, China). Antibodies against Periostin were obtained from Abmart (Shanghai, China). TRIzol reagent, PrimeScript RT Reagent Kit with gDNA Eraser, Effectene Transfection Reagent, and SYBR Premix Ex TaqII were obtained from Vazyme (Nanjing, China). SRI-011381 were obtained from MCE (Shanghai, China). The more detailed information of the antibodies was showed in Appendix A.

### 4.2. Animals and Atherosclerosis Model

*ApoE*^−/−^ mice and *Pol β*^+/−^ mice with a C57BL/6 genetic background were obtained from GemPharmatech Co., Ltd. (Nanjing, China). *ApoE*^−/−^*Pol β*^+/+^ mice and *ApoE*^+/+^*Pol β*^+/−^ mice were hybridized to obtain *ApoE*^−/−^*Pol β*^+/−^ mice. The male mice were fed a high-fat diet containing 1.25% cerholesterol (Diet D12108C; Research Diets, New Brunswick, NJ, USA) from 6 to 8 weeks of age, and the modeling was detected after 18 weeks, each group at least six mice. The animal experiments were performed in accordance with the Regulations of Experimental Animal Administrations issued by the Laboratory Animal Care Committee at Nanjing Normal University (Ethical code: IACUC-20200702).

### 4.3. Cell Culture and Stable Cell Lines

Wild-type and Pol β-knockout mouse embryonic fibroblasts (MEFs) were obtained from Professor Binghui Shen. Human coronary artery smooth muscle cells (hCASMCs) were obtained from the Otwo Biotech (Guangzhou, China) and cultured in a complete medium consisting of DMEM supplemented with penicillin (100 U/mL), streptomycin (100 μg/mL), and 15% FBS in a humidified 5% CO_2_ atmosphere at 37 °C. We used the lentivirus vector pGLV3 containing the shRNA sequence or human full-length Pol β to achieve Pol β knockout or overexpression. Oligonucleotides were showed in Appendix A.

### 4.4. Plasmid Transfection Cell Experiments

Approximately 0.3~1 × 10^5^ cells were inoculated in 500 μL complete medium 24 h before transfection, and the cell confluency was 60%~80% during transfection. Next, 0.3 μg plasmid DNA was diluted with 100 μL Opti-MEM. The solution was gently pipetted 3–5 times for mixing and was left to stand at room temperature for 5 min. The transfection reagent was gently inverted. Next, 2.0 μL of Lipofectamine TM2000 was diluted with 100 μL Opti-MEM, gently pipetted 3–5 times for mixing, and was left to stand at room temperature for 5 min. The transfection reagent and diluted plasmid DNA solution were combined and gently pipetted 3–5 times for mixing, and the solution was left to stand at room temperature for 20 min. The transfection complex was added to a 24-well cell plate at 100 μL/well, and the plate was gently shaken back and forth for adequate mixing. Cells were placed in a 37 °C, 5% CO_2_ incubator for approximately 6 h, the solution was changed to a common medium containing 10% FBS, and cells were cultured for approximately 24 h in a 37 °C, 5% CO_2_ incubator.

### 4.5. Cell Viability

Cell counting kit 8 (CCK-8) (Beyotime, Shanghai, China) was used to evaluate cell vitality. Briefly, 5 × 10^3^/well hCASMC cells were seeded into 96-well plates, and transfected with indicated plasmid. At the end of test, CCK-8 reagent was added into each well and incubated for 2 h. The absorbance values were detected by a microplate spectrophotometer at 450 nm.

### 4.6. Cell Migration and Invasion Assays

Cell migration and invasion assays were measured using wound healing and transwell assays. Regarding the wound healing assay, monolayer cells plated on a six-well plate were scratched with a pipette tip. Next, the cells were washed with PBS and cultured in a serum-free medium. The migration into the gap was imaged over several hours using a microscope.

Regarding transwell assays, the cells were dispersed into single-cell suspensions and implanted in the upper chamber of the transwell apparatus in a serum-free medium. Meanwhile, a medium with 20% FBS was added into the bottom chamber. After 24 h, the cells that passed through the polycarbonate filter were stained by 0.5% crystal violet for 15 min. The statistical results were obtained by counting nine random fields under a microscope.

### 4.7. Viral Infection and Stable Cell Line Construction

The day before infection, cells were seeded in a six-well plate at 5 × 10^5^ cells per well and 2 mL of complete medium was added. When the cell growth density reached approximately 70%, replace 1 mL of fresh complete medium. An appropriate amount of negative control virus or Pol β virus (MOI = 5) was added to each well, followed by 1 μg/mL of Polybrene auxiliary infectious agent. After the cells were cultured in the CO_2_ incubator for 48 h, the culture was continued with 2mL of fresh complete medium. After a 72 h virus infection period, the cells were observed for infection using fluorescence microscopy. Next, 2 μg/mL of puromycin was added to the culture medium for screening. After continuous screening over a period of 7 to 14 days, the cell clones were identified and expanded to establish stable cell lines expressing the gene of interest.

### 4.8. Western Blotting Analysis

Cells or tissues were homogenized in cold RIPA lysis buffer containing PMSF, and the supernatant was harvested by centrifugation. After protein quantification using a bicinchoninic acid (BCA) Protein Assay Kit, 30 µg of total protein was separated by SDS-PAGE gel and transferred to 0.22 μm PVDF membrane. The membranes were blocked using a blocking buffer (5% non-fat milk powder in TBST: 10 mM Tris-HCl pH 8.0, 150 mM NaCl, and 0.05% Tween 20) for 1 h at room temperature and incubated with primary antibodies at 4 °C overnight, followed by incubation with appropriate horseradish peroxidase-conjugated secondary antibody for 1–2 h at room temperature. Finally, they were exposed to ECL western blotting detection reagents.

### 4.9. Immunofluorescence Staining

Regarding histological analysis, tissues were fixed in 4% paraformaldehyde at 4 °C overnight, embedded in paraffin, and sliced. After deparaffinization and hydration, tissues were pretreated with sodium citrate solution for antigen retrieval. Next, the tissue sections were blocked with 5% goat serum for 30 min at room temperature and incubated with the primary antibody at 4 °C overnight, followed by incubation with the secondary antibody for 1 h at 37 °C in the dark. After counterstaining with DAPI to reveal the cell nucleus, tissue sections were photographed using a Nikon 80I 10–1500× microscope. Quantification of colocalization was analyzed by the Image J (version 1.54) colocalization finder plugin.

### 4.10. Histopathological Assessment

The tissue was excised and fixed in 4% paraformaldehyde at 4 °C overnight. After paraffin embedding, the tissue was cut into 5 μm thick slices and stained with hematoxylin-eosin. Images were obtained using a Nikon 80I 10–1500× microscope. Histopathological analysis was evaluated by independent observers in a blinded manner.

### 4.11. Oil Red O Staining

At the end of the experiment, the mice were euthanized and quantification of atherosclerotic lesions was determined using Oil Red O staining. Briefly, the entire aorta was dissected from the proximal ascending aorta to the bifurcation of the iliac artery and fixed in 4% paraformaldehyde at 4 °C overnight. After washing with PBS, the adventitia was carefully removed. The aorta was dissected longitudinally, stained with Oil Red O for 15 min at room temperature, destained for 3 min in 85% isopropanol, washed in water, and photographed using a Nikon 80I 10–1500× microscope. Regarding the aortic root, the tissue was excised after 0.9% normal saline perfusion and fixed overnight in 4% paraformaldehyde at 4 °C. Next, the tissue was embedded in paraffin, cut into 5 mm thick slices, and stained with Oil Red O using the same method. Images were obtained using microscopy.

### 4.12. Reverse Transcription-Quantitative PCR (qRT-PCR) Analysis

Total mRNA was isolated from cells and tissues using the TRIzol reagent, according to the manufacturer’s instructions, and quantified using Thermo NanoDrop 2000. Next, 1 μg of total RNA was reverse-transcribed into cDNA. qRT-PCR was performed using a reaction mixture with SYBR, according to the manufacturer’s protocols. All PCR amplifications were performed in triplicates in three independent experiments. Primers for each gene were shown in Appendix A.

### 4.13. Statistical Analysis

Statistical analysis was performed using one-way or two-way ANOVA using Prism 8.0 software (GraphPad Software 8.0, La Jolla, CA, USA). Statistical significance was set at *p* < 0.05.

## Figures and Tables

**Figure 1 ijms-25-11778-f001:**
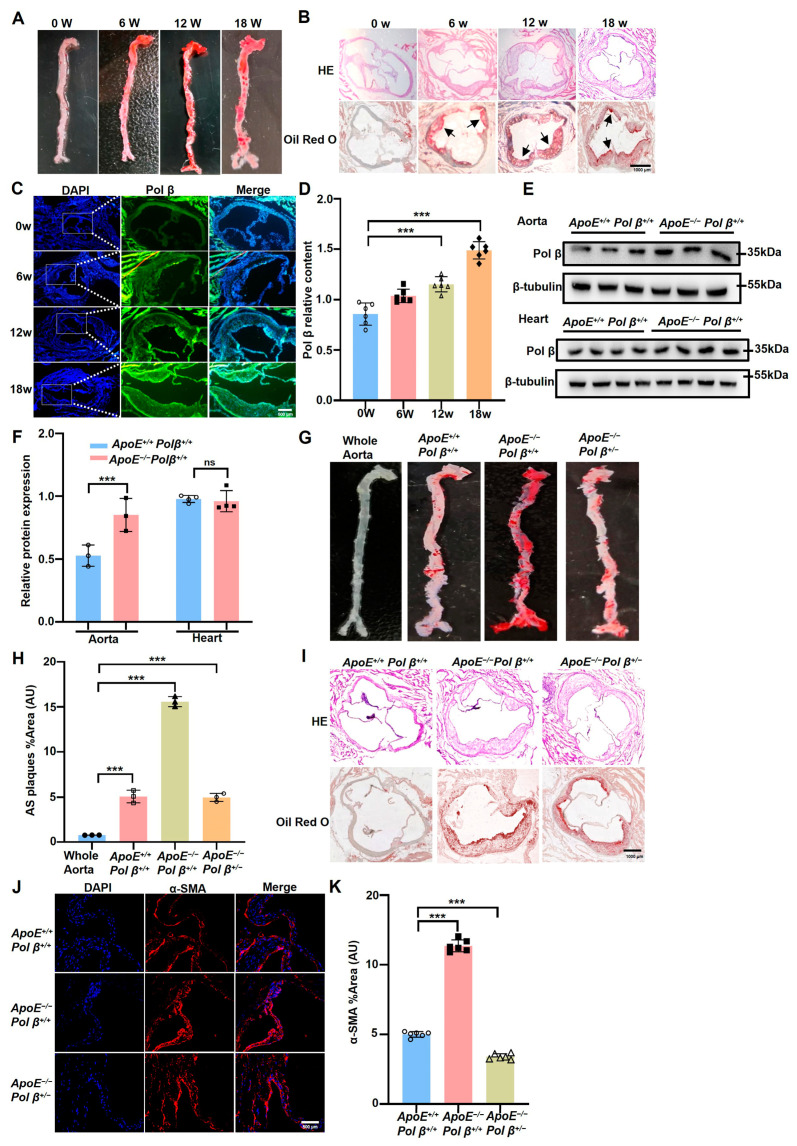
Pol β knockdown inhibits the formation of atherosclerotic plaques in *ApoE*^−/−^ mice: (**A**) Oil Red O assay was used to detect the aortic plaque formation in *ApoE*^−/−^ mice at different time points on a high-fat diet (*n =* 6 mice/treatment group). The mice were sampled after 6, 12, and 18 weeks and showed plaque progression induced by a high-fat diet; (**B**) Hematoxylin-eosin (HE) and Oil Red O staining experiments were used to observe the aortic roots of *ApoE*^−/−^ mice. The arrow marks the observed area of arterial plaque, showing the location and size of the plaque. Scale bar, 1000 μm (*n =* 6 mice/treatment); (**C**) The expression of Pol β in atherosclerotic plaques of *ApoE*^−/−^ mice was detected by immunofluorescence. The results showed that the expression of Pol β in AS plaques increased significantly; (**D**) Quantitative statistical analysis in (**C**) showed the change of Pol β expression in different treatment groups (*n =* 6 mice/treatment group); (**E**) Western blot analysis of Pol β expression in blood vessels and hearts of *ApoE*^−/−^ mice after 18 weeks of high-fat diet. The results demonstrated the differential expression of Pol β in different tissues (*n =* 3–4 mice/treatment group); (**F**) Quantitative statistical analysis of (**E**), showing the protein expression of Pol β in each treatment group (*n =* 3–4 mice/treatment group, data expressed as mean ± standard error, ^ns^, *p* ≥ 0.05; *** *p* < 0.001; Two-factor analysis of variance); (**G**) En face Oil Red O staining detected plaque formation in the aorta of *ApoE*^−/−^ mice, clearly showing lipid deposition along the aorta (*n =* 6 mice/treatment group); (**H**) Quantitative statistical analysis of (**G**), showing changes in patch area in different treatment groups (*n =* 6 mice/treatment group, data expressed as mean ± standard error, *** *p* < 0.001; One-way analysis of variance); (**I**) HE and Oil Red O staining were used to detect plaque formation in the aortic roots of *ApoE*^−/−^ mice, further illustrating the effect of Pol β on atherosclerosis. (**J**) Immunofluorescence assay was used to detect the expression of α-SMA in *ApoE*^+/+^*Pol β*^+/+^ mice, *ApoE*^−/−^*Pol β*^+/+^ mice, and *ApoE*^−/−^*Pol β*^+/−^ mice, and the experiment revealed the effect of *Pol β* knockout on key vascular markers in mice. (**K**) Quantitative statistical analysis of (**J**), showing changes in the expression of α-SMA in different treatment groups (*n =* 6 mice/treatment group, data expressed as mean ± standard error, *** *p* < 0.001; One-way analysis of variance). Scale: 1000 μm (*n =* 6 mice/treatment group, data expressed as mean ± standard error). Scale bar, 1000 μm (Data present mean ± SEM of three independent in vitro experiments; ***, *p* < 0.001; (**C**), one-way ANOVA; (**D**), two-way ANOVA; each group at least six mice).

**Figure 2 ijms-25-11778-f002:**
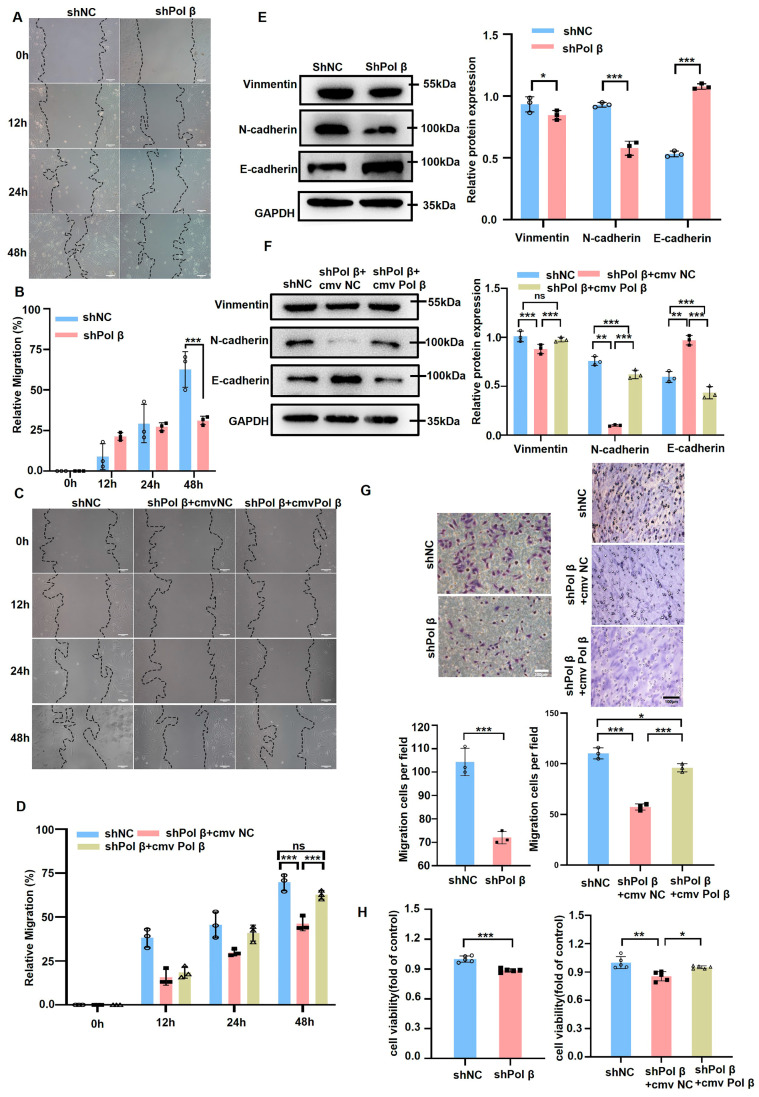
Pol β knockdown inhibits hCASMC cell migration and proliferative activity: (**A**) Cell scratch assay was used to measure the migration capacity of human coronary smooth muscle cells (hCASMC) in the control and *shPol β* groups. Scale bar, 100 μm (*n =* 3 independent experiments); (**B**) Representative images of hCASMC cell scratch tests in the control group and the shPol β group in (**A**), and a summary analysis of their migration data. Scale bar, 100 μm; (**C**) The migration of hCASMC cells was again examined by cell scratch assay to further verify the changes in cell migration ability in the shPol β group. Scale bar, 100 μm (*n =* 3 independent experiments); (**D**) Statistical analysis of the results in (**C**), showing a significant difference in hCASMC cell migration ability between the control and shPol β groups; (**E**,**F**) Western blot assay was used to detect the expression levels of migration-related proteins (such as Vimentin, N-cadherin, E-cadherin, etc.) in hCASMC cells, and the results were statistically analyzed. Experiments showed significant changes in the expression of migration-related proteins in the shPol β group (*n =* 3 independent experiments); (**G**) Transwell migration assay was used to detect and quantitatively analyze the migration ability of hCASMC cells. The results showed that the cell migration ability was significantly decreased after Pol β knockout. Scale bar, 100 μm (*n =* 3 independent experiments); (**H**) CCK-8 assay was used to detect the viability of hCASMC cells in the control group, shPol β knockdown group, and reverse Pol β knockdown group. The data showed that pol β knockdown significantly reduced cell viability, and restoration of Pol β expression partially reversed this effect (*n =* 3 independent experiments). (Data present mean ± SEM of three independent in vitro experiments; ^ns^, *p* ≥ 0.05; *, *p* < 0.05; **, *p* < 0.01; ***, *p* < 0.001; (**B**,**D**–**F**), two-way ANOVA; (**G**,**H**), one-way ANOVA).

**Figure 3 ijms-25-11778-f003:**
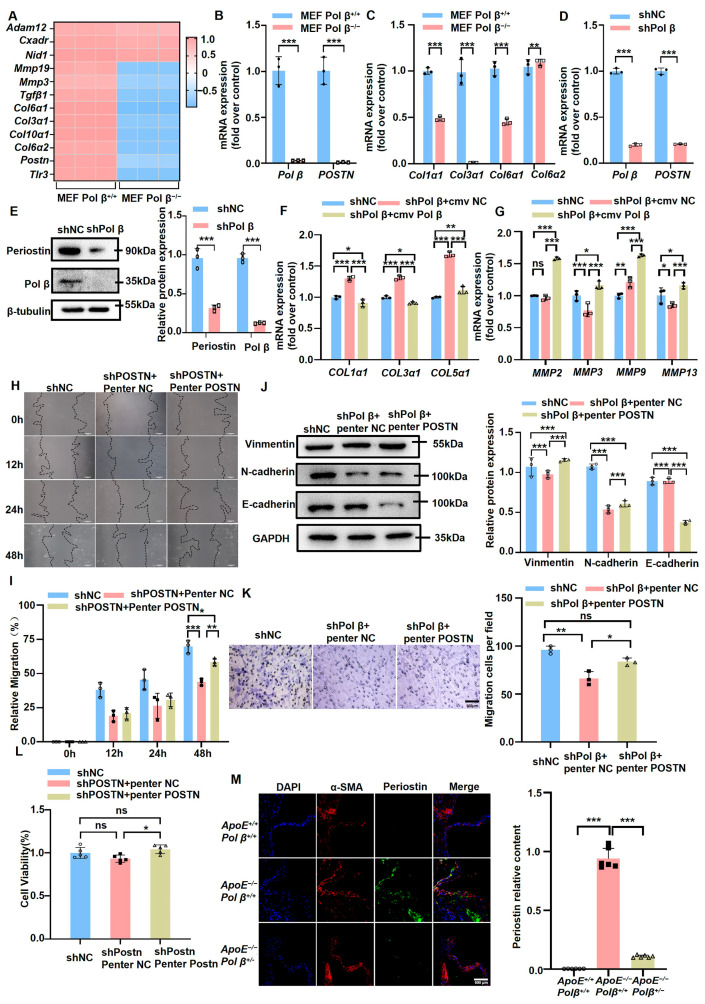
Pol β knockdown inhibits hCASMC cell migration by inhibiting the expression of periostin: (**A**) RNAseq analysis was used to detect the expression levels of multiple cytokines in MEF cells, revealing the genes whose expression changed after *Pol β* knockout (*n =* 3 independent experiments); (**B**) qRT-PCR assay was used to detect mRNA expression levels of *Pol β* and *POSTN* (periostin) in MEF cell (*n =* 3 independent experiments); (**C**) qRT-PCR assay was used to detect the expression of collagen genes in MEF cell, including *Col1α1*, *Col3α1*, etc. (*n =* 3 independent experiments); (**D**) qRT-PCR assay was used to detect the mRNA expression levels of *Pol β* and *POSTN* in hCASMCs after *Pol β* knockdown, and to verify the expression changes of matrix protein periostin (*n =* 3 independent experiments); (**E**) Western blotting assay was used to detect the expression of Pol β and periostin in hCASMCs, and quantitative statistics were performed on the data (*n =* 3 independent experiments); (**F**) qRT-PCR analysis of collagen expression levels in hCASMC cells in control group, *Pol β* knockdown group, and *Pol β* recovery group revealed the changes of collagen under different conditions (*n =* 3 independent experiments); (**G**) The expression of matrix metalloproteinases in hCASMC cells was analyzed by qRT-PCR, and the changes of matrix metalloproteinases (MMP2, MMP9, etc.) in control group, *Pol β* knockdown group, and *Pol β* recovery group were compared; (**H**) Cell scratch assay was used to detect the migration ability of hCASMC cells in control group, POSTN knockdown group, and POSTN recovery group. Scale bar, 100 μm (*n =* 3 independent experiments); (**I**) Quantitative statistical analysis of the results in (**H**), showing significant differences in hCASMC migration ability among different treatment groups (*n =* 3 independent experiments); (**J**) Western blot analysis was performed to detect the expression of proteins related to hCASMC cell migration (such as Vimentin, n-cadherin, etc.) in the control group, POSTN knockdown group, and POSTN recovery group, and quantitative statistics were performed (*n =* 3 independent experiments); (**K**) Transwell assay was used to detect the migration ability of hCASMC cells in POSTN knockdown group and POSTN recovery group, and quantitative analysis was performed, indicating that Pol β affected the migration ability of cells by regulating POSTN. Scale bar, 100 μm (*n =* 3 independent experiments); (**L**) CCK-8 assay was used to detect the viability of hCASMC cells in the control group, POSTN knockdown group, and POSTN recovery group, and the results showed that cell viability decreased after POSTN knockdown, and partially recovered after POSTN expression was restored (*n =* 5 independent experiments); (**M**) Immunofluorescence assay was used to detect the expression of α-SMA and periostin in *ApoE*^+/+^
*Pol β*^+/+^ mice, *ApoE*^−/−^*Pol β*^+/+^ mice, and *ApoE*^−/−^*Pol β*^+/−^mice, and the experiment revealed the effect of Pol β knockout on these key markers in atherosclerotic plaques. Scale bar, 500 μm (*n =* 6 mice. Data present mean ± SEM of three independent in vitro experiments; ^ns^, *p* ≥ 0.05; *, *p* < 0.05; **, *p* < 0.01; ***, *p* < 0.001; (**B**–**G**,**I**,**J**), two-way ANOVA; (**K**–**M**), one-way ANOVA).

**Figure 4 ijms-25-11778-f004:**
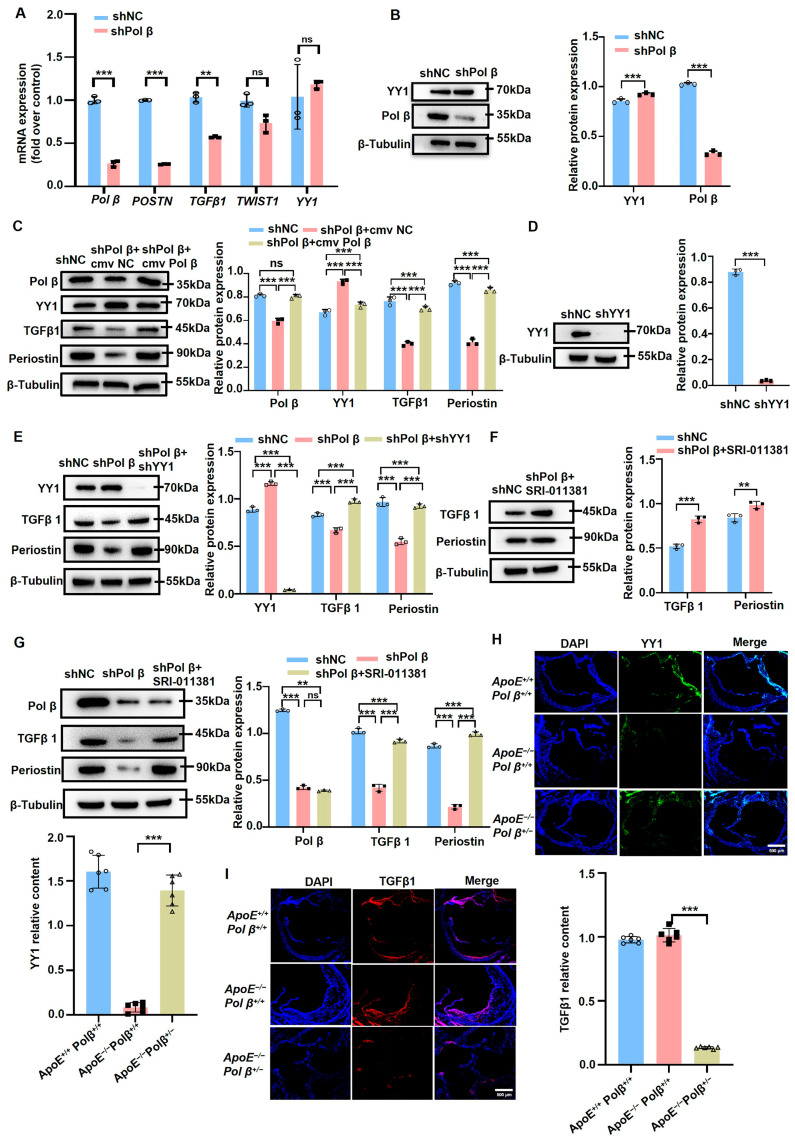
Pol β knockdown inhibits periostin expression by inhibiting YY1-TGFβ1: (**A**) mRNA expression levels of *Pol β*, *POSTN*, *TGFβ1*, *Twist1*, and *YY1* in the control group and Pol β-knockdown hCASMC cells were detected by qRT-PCR, revealing the changes in expression of these genes after *Pol β* knockdown (*n =* 3 independent experiments); (**B**) Western blot detection of YY1 protein expression in hCASMC cells after *Pol β* knockdown, and quantitative statistical analysis of the results showing the regulatory relationship between Pol β and YY1 (*n =* 3 independent experiments); (**C**–**G**) Western blotting assay was used to detect protein expression of other related proteins in hCASMC cells, including key proteins in TGFβ1 and its downstream signaling pathway. Experiments showed the effect of *Pol β* knockdown on the expression of these proteins, and the results were statistically analyzed (*n =* 3 independent experiments); (**H**,**I**) Immunofluorescence assay was used to detect the expression of TGF-β1 and YY1 at the heart base of *ApoE*^+/+^*Pol β*^+/+^ mice, *ApoE*^−/−^*Pol β*^+/+^ mice, and *ApoE*^−/−^*Pol β*^+/−^ mice, and to reveal the effect of Pol β knockdown on these markers in atherosclerosis. Scale bar, 500 μm (*n =* 6 mice, experiment included at least 6 mouse samples). Data present mean ± SEM of three independent in vitro experiments; ^ns^, *p* ≥ 0.05; **, *p* < 0.01; ***, *p* < 0.001; (**A**–**C**,**E**–**G**), two-way ANOVA; (**H**,**I**), one-way ANOVA.

**Figure 5 ijms-25-11778-f005:**
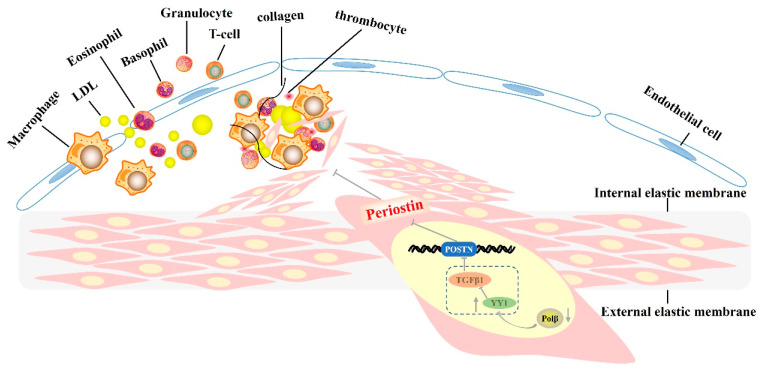
Schematic diagram showing our proposed mechanisms of Pol β deficiency inhibiting vascular smooth muscle cell migration through activating the YY1/TGF-β1/POSTN pathway.

## Data Availability

All data are available in the main text or the Appendix A.

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
