# Peer review of "Loss of DNA Polymerase β Delays Atherosclerosis in ApoE−/− Mice Due to Inhibition of Vascular Smooth Muscle Cell Migration"

_ijms, 2024, doi:10.3390/ijms252111778_

Round 1
Reviewer 1 Report
Comments and Suggestions for Authors
This manuscript by Zhao et al considers the mechanism underlying excessive migration of vascular smooth muscle cells (VSMC). This is important as VSMC migration underpins vascular conditions such as AS. Zhao studied the role of DNA polymerase β (Pol β) in VSMC migration and AS. They found increased Pol β content in AS plaques from ApoE-/-Pol β+/+ mice, and reduced SMCs viability and migration when Pol β was knocked down. Pol β knockdown inhibited POSTN gene transcription and suppression of the YY1/TGF-β1 pathway. The work is interesting but the in vivo work is still at a comparatively preliminary stage.
For example, the rationale for comparing ApoE-/-Pol β+/+ mice with ApoE+/+Pol β+/+ is not clear. It would have been more informative to quantitatively compare ApoE-/-Pol β+/+ mice with ApoE-/-Pol β+/- mice in Fig 1D/E. Also, what is the effect of Pol β deficiency on AS at the aortic sinus, arch, brachiocephalic, carotid artery, etc? Additionally, VSMC content needs to be quantified by specific stain.
Comments on the Quality of English Language
English is acceptable.
Author Response
Responses to Reviewers’ Comments
Dear Reviewer,
We would like to thank you for your critical reading of our manuscript and for providing us all the valuable suggestions. Below is a point-by-point response to your comments and suggestions with all reference lines subject to the marked version manuscript.
Comments and Suggestions for Authors:
This manuscript by Zhao et al considers the mechanism underlying excessive migration of vascular smooth muscle cells (VSMC),. This is important as VSMC migration underpins vascular conditions such as AS. Zhao studied the role of DNA polymerase β (Pol β) in VSMC migration and AS. They found increased Pol β content in AS plaques from ApoE-/-Pol β+/+ mice, and reduced SMCs viability and miaration when Pol β was knocked down. Pol β knockdown inhibited POSTN gene transcription and suppression of the YY1/TGF-β1 pathway. The work is interesting but the in vivo work is still at a comparatively preliminary stage
We really appreciate your recognition of our work and your positive comments on our findings.
Comments 1:For example, the rationale for comparing ApoE-/-Pol β+/+ mice with ApoE+/+Pol β+/+ is notclear.
Response 1:Thank you for your question, indeed, for this issue, we are not very clear in the body, because ApoE-/-Pol β+/+ mice and ApoE+/+Pol β+/+ mice were selected for comparison in order to investigate the role of Pol β in the development of atherosclerosis (AS) in the context of ApoE deletion. ApoE-/- mice are the classic model of atherosclerosis, characterized by a predisposition to arterial plaque formation on a high-fat diet. By placing the loss of function or expression of Pol β in the context of ApoE loss, it is possible to more directly observe whether and how Pol β is involved in the progression of atherosclerosis. This comparison could reveal whether Pol β has additional regulatory effects on AS during DNA damage repair, AS well as its effects on AS plaque formation and vascular smooth muscle cell migration.
Comments 2:lt would have been more informative to quantitatively compare ApoE-/-Pol β+/+ mice with ApoE-/-Pol β+/- mice in Fig 1D/E.
Response 2:Thank you for your valuable advice. According to your suggestion, we have quantitatively compared the data of ApoE-/-Pol β+/+ and ApoE-/-Pol β+/- mice in Figure 1F and 1H in the revised manuscript.
Comments 3:Also, what is the effect of Pol β deficiency on AS at the aorticsinus, arch, brachiocephalic, carotid artery, etc?
Response 3:Thank you very much for your question, which brought us more thinking. Studies have shown that Pol β deletion has a significant inhibitory effect on the progression of atherosclerosis (AS), especially in the aortic sinus, aortic arch, brachiocephalic artery, and carotid artery. Pol β is a key enzyme in the base excise repair (BER) pathway, responsible for repairing DNA damage caused by oxidative stress and other factors, which is essential to prevent the formation of AS plaques (Sobol et al., 1996; Shah & Mahmoudi, 2015). In our study, Pol β was found to be significantly upregulated in the ApoE-/- context, suggesting that it is closely associated with the development of atherosclerosis. Conversely, Polβ-deficient mice (ApoE-/-Pol β+/-) showed less plaque formation at these arterial sites, suggesting that Pol β deficiency slows the progression of AS by inhibiting the migration of vascular smooth muscle cells (VSMCs) (Schwanekamp et al., 2016).
In our study, Polβ deficiency reduces the expression of matrix protein periostin by inhibiting YY1/TGF-β1 signaling pathway, thereby inhibiting the overmigration and proliferation of VSMCs. Our study found that Pol β downregulation significantly reduced plaque formation at these critical arterial sites, particularly AS plaque reduction observed in ApoE-/-Pol β+/- mouse models (Mercer et al., 2010). These results suggest that Pol β deletion not only plays a role in DNA damage repair, but also regulates the development of AS by influencing VSMCs migration. Therefore, Pol beta may be a potential target for the treatment of atherosclerosis."
1、Sobol RW, Horton JK, Kühn R, Gu H, Singhal RK, Prasad R, Rajewsky K, Wilson SH. Requirement of mammalian DNA polymerase-beta in base-excision repair. Nature. 1996 Jan 11;379(6561):183-6. doi: 10.1038/379183a0. Erratum in: Nature 1996 Feb 29;379(6568):848. Erratum in: Nature 1996 Oct 3;383(6599):457. PMID: 8538772.
2、Shah NR, Mahmoudi M. The role of DNA damage and repair in atherosclerosis: A review. J Mol Cell Cardiol. 2015 Sep;86:147-57. doi: 10.1016/j.yjmcc.2015.07.005. Epub 2015 Jul 26. PMID: 26211712.
3、Schwanekamp JA, Lorts A, Vagnozzi RJ, Vanhoutte D, Molkentin JD. Deletion of Periostin Protects Against Atherosclerosis in Mice by Altering Inflammation and Extracellular Matrix Remodeling. Arterioscler Thromb Vasc Biol. 2016 Jan;36(1):60-8. doi: 10.1161/ATVBAHA.115.306397. Epub 2015 Nov 12. PMID: 26564821; PMCID: PMC4690815.
4、Mercer JR, Cheng KK, Figg N, Gorenne I, Mahmoudi M, Griffin J, Vidal-Puig A, Logan A, Murphy MP, Bennett M. DNA damage links mitochondrial dysfunction to atherosclerosis and the metabolic syndrome. Circ Res. 2010 Oct 15;107(8):1021-31. doi: 10.1161/CIRCRESAHA.110.218966. Epub 2010 Aug 12. Erratum in: Circ Res. 2011 Jan 7;108(1):e2. PMID: 20705925; PMCID: PMC2982998.
Comments 4:Additionally, VSMC content needs to be quantified by specific stain.
Response 4:Thank you very much for your advice. In our study, although we analyzed the migration of vascular smooth muscle cells (VSMCs) and the expression of related proteins by various experimental means such as Western blot and PCR, we believe that further quantitative analysis is necessary. In order to quantify VSMCs content more accurately, α-SMA(smooth muscle actin α chain) was selected for immunohistochemical staining. α-SMA is a marker for VSMC, and this staining allows specific labeling of VSMC and quantification of VSMC content through image analysis software. We have added the relevant results in Figure 1.
Reviewer 2 Report
Comments and Suggestions for Authors
The authors present an interesting study in which the influence of DNA polymerase beta (Pol beta) in the migration of vascular smooth muscle cells in atheroprogressive events is examined. Briefly, the authors utilised a combination of in vitro and in vivo models to demonstrate in atheroprone animals, a significant increase in Pol beta content was observed in those cells located in atheroprone areas. Cell studies which employed inhibitor and overexpression systems clearly indicated vascular smooth muscle cell migration and invasion was dictated by Pol beta levels, with knockdown linking Pol beta activity to YY1/TGF-beta signalling pathways. Taken together, these data identify Pol beta as a potentially viable therapeutic target in the treatment of atherosclerosis, and further studies are warranted in this regard.
In reviewing the manuscript I made a number of observations. The following should be considered by the authors when preparing a suitable revision.
1. The n- number should be clearly stated in each figure/figure legend. In inspecting some pieces of data, the n-number appears to fluctuate between groups e.g. Figure 1 (D). If the n-numbers were provided this could be clarified. The authors should add the n-numbers relevant to each piece of data in each figure.
2. In reviewing the supplementary material which contain the western blots, two things strike me immediately; a) most of the Westerns appear to be from sectioned blots in which only a portion of the membrane was probed, and b) in many blots there is the appearance of non-specific bands raising questions over the specificity of the antibodies being used. The authors must comment on these pints and justify the decisions derived from the Western blots.
3. In microscopy images such as Figure 1 (B) for example it would be useful if arrows were added to the images to indicate areas of interest for the reader to refer to.
4. The writing of the manuscript is good and clear for the most part, but there are a couple of typos within the piece. For example, ‘vimentin’ is written as ‘vinmentin’. The authors should revise the manuscript for instances such as these and others, and reduce their occurrence in any resubmission.
5. In reviewing some of the densitometry associated with the western blots, some values appear questionable with respect to the representative blots shown. For example, N-cadherin in Figure 2€ looks significantly lower in the ShPol beta group as compared to the ShNC group yet the data points in the densitometry suggest otherwise. The densitometry for this bar and others should be reviewed to ensure it has been performed accurately.
6. Some of the figures legends are lacking sufficient detail and only barely describe what is found in some of the figure panels. The authors must revise this aspect of the piece given many figure panels are quite busy with data, and context in most cases is sorely lacking.
7. In Figure 5, the labels need to be adjusted and made bigger in order for the reader to better interpret the pathway being displayed.
8. The concentrations of antibody used should be included in the antibody table contained in the supplementary material.
9. Were the primers listed examined to ensure they complied with MIQE guidelines standards?
Round 2
Reviewer 1 Report
Comments and Suggestions for Authors
The authors have responded adequately.
Comments on the Quality of English Language/
Author Response
Responses to Reviewers’ Comments
Dear Reviewer,
We would like to thank you for your critical reading of our manuscript and for providing us all the valuable suggestions. Below is a point-by-point response to your comments and suggestions with all reference lines subject to the marked version manuscript.
Comments and Suggestions for Authors:
The authors have responded adequately.
We are very grateful for your recognition of our work and your positive comments on our revised manuscripts. At the same time, we also proofread the full text of the returned manuscript in English again to ensure that our results can be accurately expressed.
Reviewer 2 Report
Comments and Suggestions for Authors
The authors have suitably addressed my comments for the most part, and the changes made have improved the manuscript overall. Thanks to them for engaging in the review process in such a positive way.
At the same time, I do still have concerns with regards the Western blotting, specifically the presence of the non-specific bands. Given that the blots are already cut, there is only a partial insight into how specific these antibodies are, and those areas that are probed do demonstrate non-specific bands as it is. There is a concern that the 'presumed' band at the expected molecular weight may be another protein entirely unless the appropriate measures were taken to validate such. Did the authors ever perform control experiments including recombinant forms of the target protein for example? Or use knockout samples of the target protein to demonstrate the band they are probing is the band in question?
Round 3
Reviewer 2 Report
Comments and Suggestions for Authors
The authors have suitably addressed my remaining comments.